# NAMPT and PARylation Are Involved in the Pathogenesis of Atopic Dermatitis

**DOI:** 10.3390/ijms24097992

**Published:** 2023-04-28

**Authors:** Ana B. Arroyo, Martín Bernal-Carrión, Joaquín Cantón-Sandoval, Isabel Cabas, Raúl Corbalán-Vélez, Teresa Martínez-Menchón, Belén Ferri, María L. Cayuela, Diana García-Moreno, Victoriano Mulero

**Affiliations:** 1Inmunidad, Inflamación y Cáncer, Departamento de Biología Celular e Histología, Facultad de Biología, Universidad de Murcia, 30100 Murcia, Spainicabas@um.es (I.C.); 2Instituto Murciano de Investigación Biosanitaria (IMIB)-Pascual Parrilla, 30120 Murcia, Spain; 3Centro de Investigación Biomédica en Red de Enfermedades Raras (CIBERER), Instituto de Salud Carlos III, 28029 Madrid, Spain; 4Hospital Clínico Universitario Virgen de la Arrixaca, 30120 Murcia, Spain

**Keywords:** atopic dermatitis, skin inflammation, NAMPT, PARP, NAD^+^

## Abstract

Atopic dermatitis (AD) is a chronic inflammatory skin disease of very high prevalence, especially in childhood, with no specific treatment or cure. As its pathogenesis is complex, multifactorial and not fully understood, further research is needed to increase knowledge and develop new targeted therapies. We have recently demonstrated the critical role of NAD^+^ and poly (ADP-ribose) (PAR) metabolism in oxidative stress and skin inflammation. Specifically, we found that hyperactivation of PARP1 in response to DNA damage induced by reactive oxygen species, and fueled by NAMPT-derived NAD^+^, mediated inflammation through parthanatos cell death in zebrafish and human organotypic 3D skin models of psoriasis. Furthermore, the aberrant induction of NAMPT and PARP activity was observed in the lesional skin of psoriasis patients, supporting the role of these signaling pathways in psoriasis and pointing to NAMPT and PARP1 as potential novel therapeutic targets in treating skin inflammatory disorders. In the present work, we report, for the first time, altered NAD^+^ and PAR metabolism in the skin of AD patients and a strong correlation between NAMPT and PARP1 expression and the lesional status of AD. Furthermore, using a human 3D organotypic skin model of AD, we demonstrate that the pharmacological inhibition of NAMPT and PARP reduces pathology-associated biomarkers. These results help to understand the complexity of AD and reveal new potential treatments for AD patients.

## 1. Introduction

Atopic dermatitis (AD) is a chronic inflammatory skin disease of very high prevalence, especially in childhood, which has no specific treatment or cure [1]. Epidermal barrier dysfunction and dysregulation of the Th2 immune response play a crucial role in the pathogenesis of this disease, which can be further complicated by other genetic and environmental factors [1]. As its pathogenesis is complex, multifactorial and not fully understood, further research is needed to increase knowledge and develop new targeted therapies.

Nicotinamide adenine dinucleotide (NAD^+^) is essential in cell redox reactions as a hydrogen carrier. It is involved in multiple vital cellular processes, such as mitochondrial function, the immune response, inflammation and DNA repair [2]. The Preiss-Handler, de novo and salvage pathways are responsible for tightly regulated NAD^+^ biosynthesis. Dietary niacin (vitamin B3) is the main NAD^+^ precursor, consisting of nicotinamide (NAM), nicotinic acid (NA) and nicotinamide riboside (NR) [2]. Many enzymes consume NAD^+^ to perform their functions by releasing NAM, which allows the NAD^+^ pool to be maintained in many tissues through the NAD^+^ salvage pathway [3]. Nicotinamide phosphoribosyltransferase (NAMPT), the rate-limiting enzyme in the NAD^+^ salvage pathway, transforms NAM into nicotinamide mononucleotide (NMN), which in turn is converted into NAD^+^ by NMN adenylyltransferases (NMNAT 1–3) [2,3]. Interestingly, NAMPT has been associated with different inflammatory conditions [4], including psoriasis (PS), another chronic inflammatory skin disease [5,6]. Moreover, progressive NAD^+^ depletion caused by the pharmacological inhibition of NAMPT with FK-866, a specific non-competitive inhibitor [7], has shown anti-inflammatory effects in different experimental settings [8,9].

Poly (ADP-ribose) (PAR) polymerases (PARP) are the main NAD^+^-consuming enzymes during the ADP-ribosylation or PARylation process, which involves the transfer of ADP-ribose molecules (linear or branched PAR) to target proteins [10]. The PARP family contains at least 17 members in humans and is divided into four subfamilies according to the functional domain. PARP1, the most intensively and extensively studied member, is primarily responsible for PAR biosynthesis and DNA damage repair [10]. Beyond its role in DNA damage repair, PARylation is also involved in inflammation, metabolism and cell death [11]. Different stimuli can cause the overactivation of PARP1, resulting in excessive PARylation with the accumulation of PAR residues, triggering a specific type of cell death called parthanatos. Parthanatos is a multistep cascade involving PAR binding to Apoptosis-Inducing Factor Mitochondria Associated 1 (AIFM1), the release of AIFM1 from mitochondria, the nuclear translocation of the AIFM1/macrophage Migration Inhibitory Factor (MIF) complex, and MIF-mediated large-scale DNA fragmentation [12]. To date, several inhibitors of PARP enzyme activity, such as Olaparib, have been developed for the treatment of various types of cancer [13].

Recently, our group has demonstrated a critical role of NAD^+^ and PAR metabolism in oxidative stress and skin inflammation [5]. Specifically, we found that hyperactivation of PARP1 in response to reactive oxygen species (ROS)-induced DNA damage, and fueled by NAMPT-derived NAD^+^, mediated inflammation through parthanatos cell death in preclinical zebrafish and human organotypic 3D skin models of PS [5]. Clinical data also support the role of these signaling pathways in PS, pointing to NAMPT and PARP1 as potential new therapeutic targets in treating inflammatory skin disorders [5].

In this work, we report, for the first time, the alteration of NAD^+^ and PAR metabolism in the skin of AD patients. Furthermore, using a human 3D organotypic skin model of AD, we demonstrate the benefit of the pharmacological inhibition of NAMPT and PARP1, which reduce AD-associated inflammation and keratinocyte proliferation. In doing so, we extend the understanding of this complex disease and reveal potential new treatments for AD patients.

## 2. Results

### 2.1. NAMPT and PARylation Increase in Human AD Lesions

To determine whether NAD^+^ metabolism is involved in the cutaneous manifestation of AD, we analyzed the levels of NAMPT and PARylated protein in skin biopsies from AD patients and healthy individuals by immunohistochemistry (Figure 1). The results showed that NAMPT was hardly detected in the healthy epidermis and dermis (Figure 1A,B). However, NAMPT was widely overexpressed in the spinous layer and in a few basal keratinocytes and dermal cells in AD skin (Figure 1A,B). Interestingly, NAMPT immunoreactivity was mainly found in the nuclei of keratinocytes, but weaker immunoreactivity was also observed in their cytoplasm (Figure 1A,B). Consistently, although PAR immunoreactivity was found in the nuclei of scattered keratinocytes of the spinous layer from healthy subjects, it was widely observed in the nuclei of most keratinocytes of the spinous layer and dermal fibroblasts of AD skin (Figure 1C,D), also demonstrating increased PARylation in lesional skin from AD patients.

### 2.2. The Expression of Genes Encoding NAD^+^ and PAR Metabolic Enzymes Is Dysregulated in AD

We next performed a detailed study of the expression of genes encoding enzymes of the NAD^+^ and PAR metabolic pathways in AD. For this purpose, we analyzed transcriptomic data from two different cohorts of human AD included in the GEO database. Cohort 1 (GSE57225) compared patients affected by atopic eczema with healthy skin. In addition, cohort 2 (GSE32924) included an analysis of matched samples of non-lesional AD (ANL) and lesional AD (AL) compared to skin of thelathy individuals. Interestingly, transcriptomic analysis of both human AD cohorts revealed a differential expression profile of genes encoding enzymes involved in NAD^+^ (Figure 2) and PAR metabolism (Figure 3), being more marked in subjects with AL. First, we explored the three canonical pathways for the synthesis of NAD^+^: Preiss-Handler, the de novo biosynthesis pathway and the salvage pathway (Appendix A). We found that the transcript levels of genes encoding several enzymes involved in this process were increased in AD disease (Figure 2A). Notably, the levels of NAMPT, the limiting enzyme of this pathway, increased in patients with AD compared to controls. Likewise, the Preiss-Handler pathway appears to be more active in AD skin, as the increased expression of genes encoding nicotinate phosphoribosyl transferase (NAPRT) and NAD synthetase (NADSYN), two key enzymes in this pathway, was observed (Figure 2B,F). In addition, genes encoding indoleamine 2,3-dioxygenase 1 (IDO1) and tryptophan 2,3-dioxygenase (TDO2), which are involved in the first limiting step of the de novo pathway, were also induced in AD compared to healthy skin (Figure 2C,G). Considering that NAD^+^ is a balance between its consumption and synthesis, we also examined the transcript levels of genes encoding enzymes that consume or degrade NAD^+^, such as CD38 and PARP (Appendix A), which were also found to be elevated in AD skin (Figure 2D and Figure 3A,C). We next analyzed the transcript levels of several genes encoding PARP family members, AIFM1 and MIF, which are indispensable parthanatos components, as well as those encoding different PAR hydrolases that negatively regulate protein PARylation (Appendix A). Transcriptomic data revealed strongly increased mRNA levels of *PARP1*, *AIFM1* and *MIF* in AD (Figure 3A,C). Furthermore, although no alteration was found in the expression profile of the gene encoding PARG, the transcript levels of genes encoding several PAR hydrolases, namely MACROD1, MACROD2, TARG, ENPP1 and NUDT16, were lower in AD (Figure 3B,D). These results, taken together, indicate that AD may show increased PARylation and, therefore, may require elevated NAD^+^ synthesis due to its exacerbated consumption.

### 2.3. NAMPT and PARP1 Expression Correlated with the Lesional Status of AD

Since the highest differential expression of NAD^+^ and PAR metabolic pathways was found in AL compared to ANL (Figure 2E–G and Figure 3C,D), we also tested whether there might be a correlation between these alterations and AD lesional status. First, we observed a strong positive correlation between the transcript levels of *NAMPT* and *PARP1* genes using the GEO cohort 2 database (Figure 4A). Next, we correlated *NAMPT* and *PARP1* transcript levels with biomarkers of epidermal hyperplasia (*K16*, *Ki67* and *PCNA*) (Figure 4A), type 2 inflammation (*CCL17*, *CCL18*) and other inflammatory biomarkers (*MMP9* and CA2) (Figure 4B) that have already been studied in AL and ANL samples [14,15]. Interestingly, we found a positive correlation between *NAMPT* and *PARP1* transcript levels and those of the different inflammatory and proliferative biomarkers assessed (Figure 4A,B). In concordance, the immunohistochemical analysis of AD patients showed that patients with elevated NAMPT and PAR levels (Figure 1) also had an elevated signal of the proliferation marker PCNA in the basal and spinous layer (Figure 4C). Notably, double immunofluorescence analysis revealed that most NAMPT^+^ cells were in the spinous layer, while most PCNA^+^ cells were found in the basal layer. In addition, the morphological analysis confirmed that there was not statistically significant colocalization (Appendix A).

### 2.4. Pharmacological Inhibition of NAMPT and PARP1 Decreased the Expression of Pathology-Associated Biomarkers in Human Organotypic 3D Skin Model of AD

Given the results obtained in patients with AD, we decided to study the impact of the pharmacological inhibition of NAMPT and PARP1 using a human organotypic 3D skin model of AD (Figure 5A). As expected, following the addition of AD-associated cytokines (Th2 cytokines IL4 and IL13) to the 3D skin culture, we found increased transcript levels of the inflammatory markers *CA2*, *NELL2* and *CCL26* accompanied by a decrease in the epidermal differentiation markers *FLG* and *LOR* and increased expression of the proliferation marker *PCNA* (Figure 5B–G). All these markers used to characterize the human organotypic 3D skin model of AD were validated in the two cohorts of AD patients (Appendix A). Interestingly, after the pharmacological inhibition of NAMPT and PARP1, we observed a decrease in inflammatory and proliferation markers (Figure 5B–D,G). Surprisingly, the downregulation of *CA2* and *NELL2* expression was completely abrogated when we inhibited both enzymes at the same time (Figure 5B–D). Furthermore, and unexpectedly, only treatment with the PARP1 inhibitor Olaparib was able to discretely recover the differentiation markers *FLG* and *LOR* (Figure 5E,F). It is also important to note that although NAMPT was not elevated after AD cytokine stimulation, PARylation was drastically induced and, more importantly, completely reversed by the inhibition of NAMPT or PARP (Figure 5I,J). These results confirm that the inhibition of both NAMPT and PARP alleviated inflammation in this AD model.

## 3. Discussion

Despite the high frequency and significant burden of AD in both pediatric and adult populations, its pathogenesis has not yet been fully characterized and therapeutic resources remain limited [16]. A better understanding of the molecular basis of AD would have important implications for the development of new therapeutic drugs. Here, we demonstrate for the first time that NAD^+^ and PAR metabolism are involved in the pathogenesis of AD (Figure 6). An analysis of human transcriptomic data from AD skin showed altered expression profiles of genes encoding NAD^+^ metabolic enzymes, including the salvage, Preiss-Handler and de novo pathways, and of genes encoding key enzymes involved in PAR metabolism and parthanatos. NAD^+^ synthesis is required for PARylation; considering the transcriptomic data, all NAD^+^ synthesis pathways are increased, which could indicate that there is increased consumption and demand for NAD^+^ in AD. Furthermore, PAR homeostasis is a balance between PAR formation by members of the PARP superfamily and its degradation catalyzed by PAR hydrolases [17]. Our data showed not only that PARP family members are overexpressed in AD, but that several PAR-degrading enzymes are also downregulated. Therefore, the hyper-PARylation that we found in AD skin biopsies could be a consequence of PAR overproduction as well as insufficient degradation. In addition, AD skin is characterized by significant cellular immune infiltration and genes whose expression is altered in lesional skin could be expressed in tissues other than keratinocytes. However, our immunohistochemical analysis of AD lesional skin confirmed the drastic induction of NAMPT at the protein level and the accumulation of PAR in the nuclei of epidermal keratinocytes and dermal cells. Overall, these findings are similar and may be reflecting the same mechanisms that we described in PS [5], where preclinical zebrafish models and a human organotypic 3D skin model of PS supported that NAD^+^ synthesis was induced to meet the PARylated demand, leading to death by parthanatos [5]. Curiously, we did not find colocalization of NAMPT and PCNA in AD lesional skin. It is likely, therefore, that NAMPT^+^ keratinocytes release NAD^+^, as has been shown in a mouse model of acute inflammation [18,19], to sustain the high proliferation rate in the whole epidermis.

Importantly, changes in the transcriptomic profile of AD have been described depending on the lesional state [14] and appear to be useful in monitoring the response to treatment [15]. Here, we found that the alterations in key genes involved in both NAD^+^ synthesis and PAR metabolism are more drastic in AL. Furthermore, correlation studies showed an association between NAMPT and PARP1 with crucial markers of inflammatory and hyperplasia in AL. Therefore, these findings are very interesting, since they suggest the importance of these proteins in the course or progression of lesions.

In addition to its enzymatic activity in NAD^+^ synthesis, there is an extracellular form of NAMPT (eNAMPT, also called visfatin) that can be secreted by different cell types, such as adipocytes [20]. eNAMPT is considered a proinflammatory adipokine as it promotes the release of inflammatory cytokines in immune cells [21] and keratinocytes [22], among others. In addition, it has been associated as a biomarker in the serum of different inflammatory pathologies, including PS [23]. In this line, there are only two studies regarding the involvement of eNAMPT in AD. In 2013, Suga et al. identified increased serum levels of eNAMPT in adult AD patients compared to healthy controls. They also found that the serum levels of NAMPT in these patients correlated with other disease markers, such as the eosinophil count. In addition, increased expression of NAMPT was detected in the adipose tissue of skin with AD lesions [24]. However, the opposite was observed in children with AD, who had reduced levels of serum eNAMPT [25]. On this point, the authors suggested that there might be differences in the underlying mechanisms of childhood and adult AD, but also in the timing of sample collection. Although further studies are needed, eNAMPT levels could be a potential serum biomarker of adult AD. Future studies analyzing serum and skin levels in the same patient, as well as their variations in response to treatment, would be very interesting to determine their clinical value.

Several 3D skin models capable of mimicking AD pathology have been developed and have been very useful in drug testing and in the approach to this disease [26]. One of the main differences between the available models is the inclusion or not of immune cells. Interestingly, in co-culture models, it has been shown that drugs that act directly on inflammation, such as dexamethasone or cyclosporin A, are able to suppress inflammatory cell infiltrates by inhibiting the cytokine production triggered by T cells present in the model. However, with these drugs, no improvement in epidermal spongiosis or in the expression of specific AD inflammatory markers CA2 and NELL2 was observed in a model lacking immune cells [27]. Therefore, it would be more appropriate to test other treatments specifically targeting epidermal characteristics in models lacking immune cells. In our study, we focused on specifically abolishing NAD^+^ and PAR metabolism in keratinocytes. Inhibition of PARP1 and NAMPT activity by Olaparib and FK-866, respectively, reversed Th2 cytokine-induced epidermal and inflammatory skin damage in our human organotypic 3D skin model of AD. Unexpectedly, we found that, unlike in patients, NAMPT was not induced in the 3D model, which could be due to the timing and dose of Th2 cytokines used to induce AD. However, pharmacological inhibition of NAMPT is beneficial in this model, which could be due to its indirect effect on PARylation, which is highly induced, although the benefit is much more marked when both inhibitors are combined. In our view, our results have added value, as most available treatments are based on the inhibition of the immune response [28] and do not target the aberrant keratinocyte activation that may be key in the pathology.

Furthermore, it is well known that AD is associated with other atopic comorbidities, particularly food allergies, asthma and allergic rhinitis [29], which adds complexity to the management of the disease. Indeed, many studies have shown that PARP enzymes play key roles in the regulation and progression of the inflammatory processes in asthma and allergic rhinitis [30]. Consequently, the inhibition of PARP by genetic ablation or the use of pharmacological agents in different animal models of asthma has demonstrated therapeutic effects against lungs acting with a specific effect on immune cell recruitment and modulating the production of asthma-associated cytokine [30]. Therefore, PARylation seems to be especially relevant in AD, as we detected many PAR molecules both in the skin of patients and in the 3D model. Therefore, the benefit of PARP inhibitors might be greater in AD patients with associated comorbidities.

The idea of repositioning PARP inhibitors in non-oncological diseases has been intensively considered [31], especially considering that PARP inhibitors may be therapeutically advantageous for a variety of non-oncological indications that are supported by an adequate risk/benefit analysis [31]. Here, we shed light on the potential new use of these drugs in AD, opening the door for future in-depth explorations through preclinical studies of the mechanisms involved, as well as the safety and efficacy of this approach.

## 4. Methods and Materials

### 4.1. Immunohistochemistry on Human Skin Samples

Skin biopsies from healthy donors (*n* = 10) and AD patients (*n* = 6) were used. Clinical inclusion criteria were an AD diagnosis evaluated by 2 dermatologists independently and histology of skin lesions compatible with spongiotic dermatitis evaluated by a pathologist. Clinical exclusion criteria were a diagnosis compatible with other dermatoses (lichen planus, phytodermatosis, allergic contact eczema, psoriasis/psoriasiform dermatitis) and histology compatible with lichenoid dermatitis/lichen planus or psoriasiform dermatitis. Sections were fixed in 4% PFA, embedded in Paraplast Plus and sectioned at a thickness of 5 μm. After being dewaxed and rehydrated, the sections were incubated in 10 mM citrate buffer (pH 6) at 95 °C for 30 min and then at room temperature for 20 min to retrieve the antigen. Next, steps were taken to block endogenous peroxidase activity and nonspecific binding. The sections were then immunostained with mouse monoclonal antibodies to NAMPT (sc-166946, Santa Cruz Biotechnology, 1/100), poly (ADP-ribose) (ALX-804-220, Enzo Life Sciences, Madrid, Spain, 1/100) and PCNA (sc-25280, Santa Cruz Biotechnology, Heidelberg, Germany, 1/100), followed by 1/100 dilution of biotinylated secondary antibody and then by the application of the ImmunoCruz goat ABC Staining System (sc-2023, Santa Cruz Biotechnology). Finally, after adding DAB staining solution, the sections were dehydrated, rinsed and mounted in Neo-Mount. No staining was observed when the primary antibody was omitted (Appendix A). Sections stained with DAB were finally examined under a Leica microscope equipped with a Leica DFC 280 digital camera, and photographs were processed with the Leica QWin Pro software 3.1.0. Image analysis was performed by selecting the epidermis area as the region of interest (ROI) with the ImageJ software 1.54d (FIJI).

For the double immunofluorescence analysis of NAMPT and PCNA, skin human biopsies processed as above were treated for 20 min at 95 °C with Tris/EDTA buffer (10 mM Tris base, 1 mM EDTA solution, 0.05% Tween 20, pH 9.0) for antigen retrieval, blocked with 5% BSA in PBS for 30 min at RT and washed with PBS. Double immunostaining was performed with rabbit anti-NAMPT (1:1000, ab236874, Abcam, Cambridge, UK) and mouse monoclonal anti-PCNA (1:1000, P8825, Sigma-Aldrich, Burlington, MA, USA) antibodies, followed by secondary antibodies donkey anti-rabbit IgG H&L Alexa Fluor^®^ 488 (# A21206) and goat anti-mouse IgG H&L Alexa Fluor^®^ 594 (#A11032), both from Thermo Fisher Scientific (Waltham, MA, USA) and used at 1:400 in incubation buffer (PBS + 1% BSA + 0.3% Triton X-100). Sections were then washed with PBS and incubated with DAPI (1:5000; Sigma-Aldrich) for 5–7 min in darkness and at room temperature. Finally, sections were mounted using VECTASHIELD^®^ Antifade Mounting Medium with DAPI (Vector Laboratories, H-1200) and examined under a confocal microscope (Leica STELLARIS 8) and further processed with the Leica software and ImageJ 1.54d. No staining was observed when primary antibodies were omitted. The objective used was x63. The correlation of NAMPT and PCNA fluorescence intensity was determined using ImageJ and the JACoP plugin (Lachmanovich method), and specifically the Pearson correlation coefficient and Manders (M1 and M2) coefficients. In the Pearson correlation test, the R value ranges between −1 and 1, with 1 being a perfect correlation, 0 no linear correlation and −1 a perfect negative linear correlation. The M1 and M2 coefficients indicate the overlapping of both fluorescences, where 0 indicates no overlapping and 1 the maximum overlap. M1 denotes the overlapping of NAMPT with PCNA. M2 denotes the overlapping of PCNA with NAMPT.

### 4.2. Gene Expression Omnibus (GEO) Database

Human AD transcriptomic data were collected from two different cohorts (accession numbers: GSE57225 and GSE32924) in the GEO database (https://www.ncbi.nlm.nih.gov/geo/, accessed 12 December 2022).

The GSE32924 cohort includes an analysis of paired samples of non-lesional AD (ANL) and lesional AD (AL) skin biopsies compared to normal skin. ANL skin is characterized by terminal differentiation defects. The GS357225 cohort includes patients affected by both PS and non-atopic or atopic eczema, as well as healthy skin. Here, we have used only data from patients with AD and controls. Gene expression graphs were obtained using the GraphPad Prism software 8.2.1.441.

### 4.3. Human Organotypic 3D Skin Model of AD

Insert transwells (Merck, Rahway, NJ, USA, MCHT12H48) were seeded with 200,000 human foreskin keratinocytes (Ker-CT, ATCC CRL-4048) on the transwells in 300 μL CnT-PR medium (CellnTec) in a 12-well format. After 48 h, the cultures were switched to CnT-PR-3D medium (CELLnTEC, Bern, Switzerland) for 24 h and then cultured in the air–liquid interface for 17 days. From day 12 to 17 of the air–liquid interphase culture, Th2 cytokines IL13 (10 ng/mL) and IL4 (10 ng/mL) (PeproTech, London, UK) were added [32]. Pharmacological treatments were applied from day 14 to 17 and consisted of 100 μM Olaparib, 100 μM FK-866 or their combination. The culture medium was refreshed every 2 days. On day 17, tissues were harvested for protein and gene expression analysis.

### 4.4. Western Blot

The 3D cell cultures were lysed in 0.1 mL of whole-cell extract (WCE) buffer [50 mM Tris-HCl pH 8.0, 280 mM NaCl, 0.5% IGEPAL, 10% glycerol and 5 mM MgCl_2_] supplemented with phosphatase, protease and poly (ADP-Ribose) glycohydrolase (PARG) inhibitors, where appropriate. The extract was then incubated for 1 h at 4 °C, and the cell debris was removed by centrifugation at 20,000× *g* for 1 h at 4 °C. The Pierce BCA Protein Assay Kit (Thermo Scientific, Waltham, MA, USA) was used to quantify proteins. Cell lysates (30 µg) were then mixed with 5X protein-loading buffer, and samples were subjected to electrophoresis on a polyacrylamide gel and transferred to PVDF membranes. The membranes were incubated for 1 h with Tris-buffered saline with Tween 20, pH 8.0 (TTBS), containing 5% (*w*/*v*) BSA (BSA-TTBS). The membranes were incubated overnight at 4 °C with antibodies to human PAR (ALX-804-220, 1/400), NAMPT (sc-166946, 1/100) and β-actin (ACTB, sc-47778, 1/1000). After washing, the membranes were incubated with an appropriate HRP-conjugated secondary antibody diluted 1:2500 in BSA-TTBS. ChemiDoc XRS BioRad enhanced chemiluminescence reagent and ChemiDoc were used for protein detection. Densitometric analyses were performed using the ImageJ software and normalized to the ACTB band intensity.

### 4.5. Analysis of Gene Expression

Total RNA from 3D cell cultures was isolated using TRIzol reagent and the RNAqueous Micro Kit (Thermo Fisher Scientific) according to the manufacturer’s protocol. The SuperScript VILO cDNA Synthesis Kit (Thermo Fisher Scientific) was used to synthesize first-strand cDNA from 1 µg of total RNA, pre- treated with DNase I (1 U/mg RNA, Thermo Fisher Scientific). Quantitative real-time PCR (qPCR) was performed with an ABI PRISM 7500 instrument (Thermo Fisher Scientific) using Power SYBR Green Master Mix (Applied Biosystems, Waltham, MA, USA). All primers (Appendix A) were obtained from Merck. Gene expression was normalized to the endogenous *ACTB* content in each sample following the Pfaffl method [33]. qPCR was performed in triplicate for each sample and from 3 independent experiments.

### 4.6. Statistical Analysis

Data were analyzed by analysis of variance (ANOVA) and a Tukey multiple range test to determine differences between groups with a Gaussian data distribution. Nonparametric data were analyzed by the Kruskal–Wallis test and Dunn’s multiple comparisons test. Differences between 2 samples were analyzed using Student’s *t*-test or the nonparametric Mann–Whitney test, as appropriate. Univariate correlations were calculated using the Spearman’s correlation coefficient. Data are shown as mean ± SEM.

## 5. Conclusions

We have shown that NAD^+^ and PAR metabolism are involved in the pathogenesis of AD. PARP1 hyperactivation in keratinocytes, likely fueled by NAD^+^ derived from NAMPT, exacerbates AD inflammation and hyperplasia. Accordingly, the pharmacological inhibition of NAMPT and PARP1 decreases disease-associated inflammation and proliferation, restoring skin homeostasis. Taken together, as we have previously described in other inflammatory skin disorders, such as PS, our results point to these drugs as potential novel therapeutic targets to treat this disease.

## Figures and Tables

**Figure 1 ijms-24-07992-f001:**
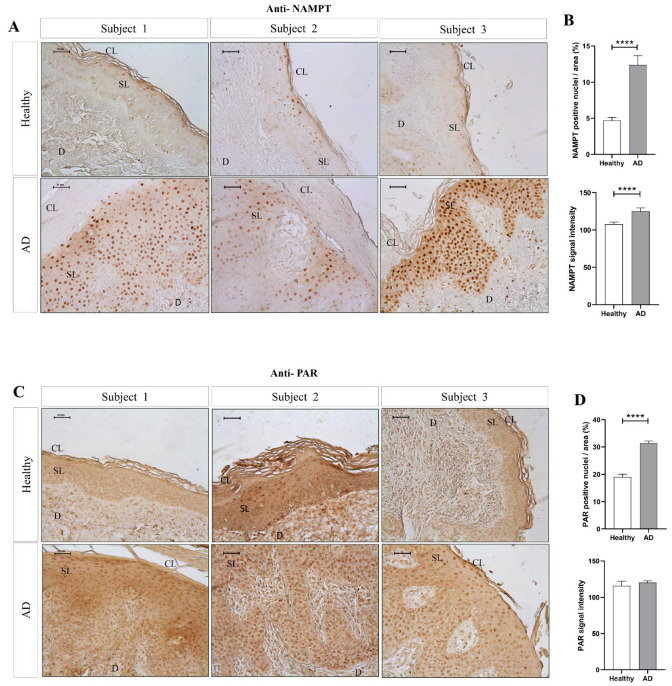
The amount of NAMPT protein and PARylation increased in lesional skin of AD patients. Representative images and analysis of biopsy sections from healthy (*n* = 10) and AD (*n* = 6) skin biopsies immunostained with anti-NAMPT (**A**,**B**) and anti-PAR (**C**,**D**). The mean ± SEM of each group is shown. *p* values were calculated using the nonparametric Mann–Whitney test, **** *p* ≤ 0.0001. Scale bar is 50 μm in all panels. CL: cornified layer; D: dermis; SL: spinous layer.

**Figure 2 ijms-24-07992-f002:**
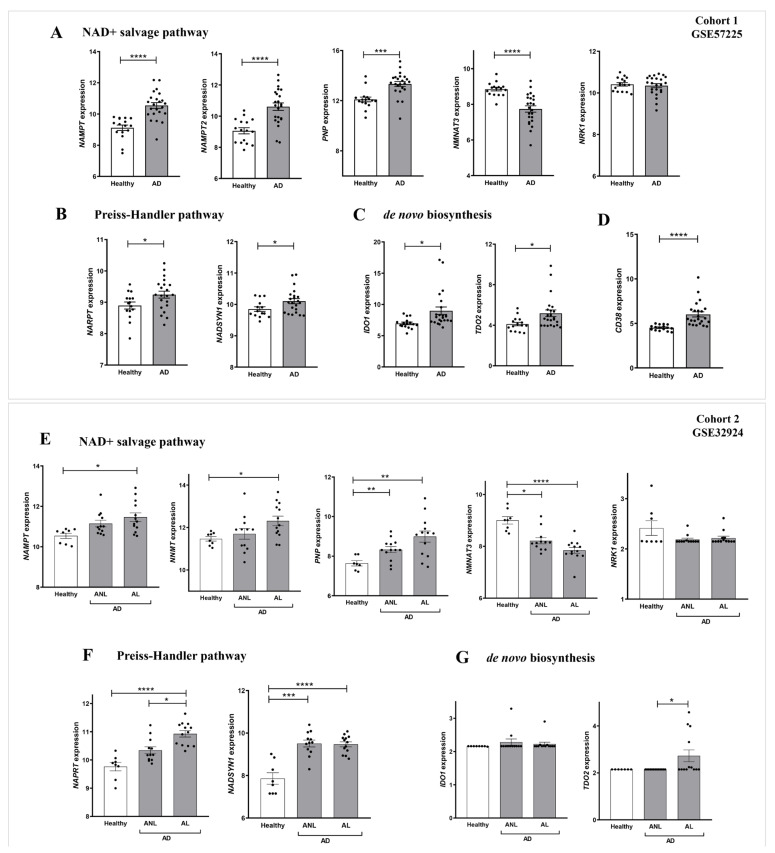
Differential expression profiles of genes encoding key NAD^+^ metabolic enzymes in AD. Transcriptomic data obtained from two different human AD cohorts, GSE57225 (**A**–**D**) and GDS4491 (**E**–**G**), from the GEO database. In cohort 1, AD was compared to healthy skin samples (**A**–**D**), and in cohort 2, skin with lesional (AL) and non-lesional (ANL) AD was compared to healthy skin samples (**E**–**G**). Each point represents one individual and the mean ± SEM for each group is also shown. *p* values were calculated using unpaired Student’s *t*-test, one-way ANOVA and Tukey’s multiple range, as appropriate. * *p* ≤ 0.05, ** *p* ≤ 0.01, *** *p* ≤ 0.001, **** *p* ≤ 0.0001.

**Figure 3 ijms-24-07992-f003:**
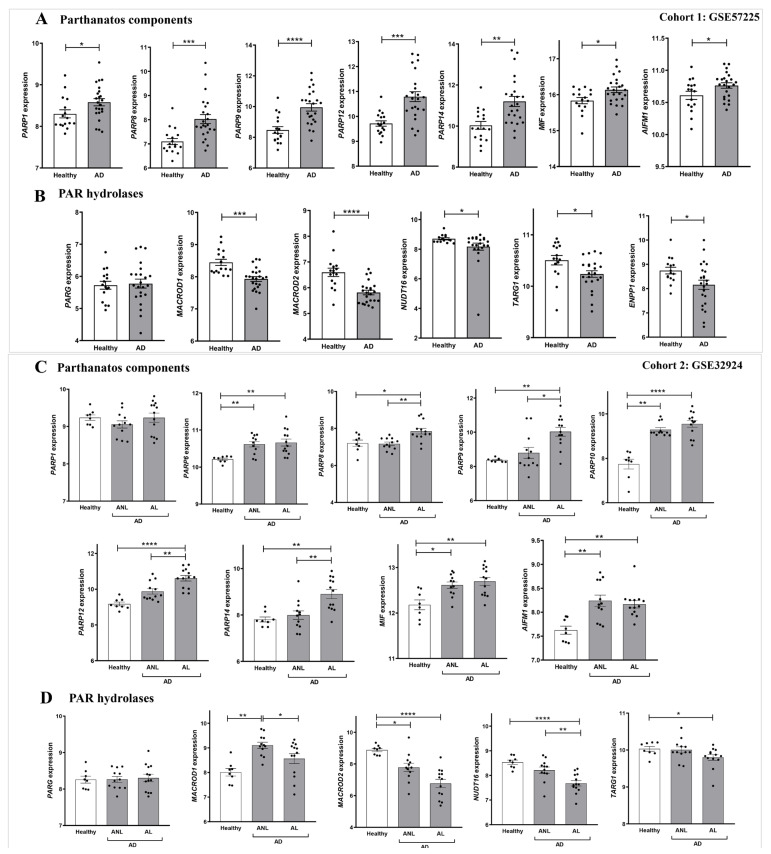
Differential expression profiles of genes encoding parthanatos components in AD. Transcriptomic data obtained from two different human AD cohorts, GSE57225 (**A**,**B**) and GDS4491 (**C**,**D**), from the GEO database. In cohort 1, AD was compared to healthy skin samples (**A**,**B**), and in cohort 2, skin with lesional (AL) and non-lesional (ANL) AD was compared to healthy skin samples (**C**,**D**). Each point represents one individual and the mean ± SEM for each group is also shown. *p* values were calculated using unpaired Student’s *t*-test, one-way ANOVA and Tukey’s multiple range, as appropriate. * *p* ≤ 0.05, ** *p* ≤ 0.01, *** *p* ≤ 0.001, **** *p* ≤ 0.0001.

**Figure 4 ijms-24-07992-f004:**
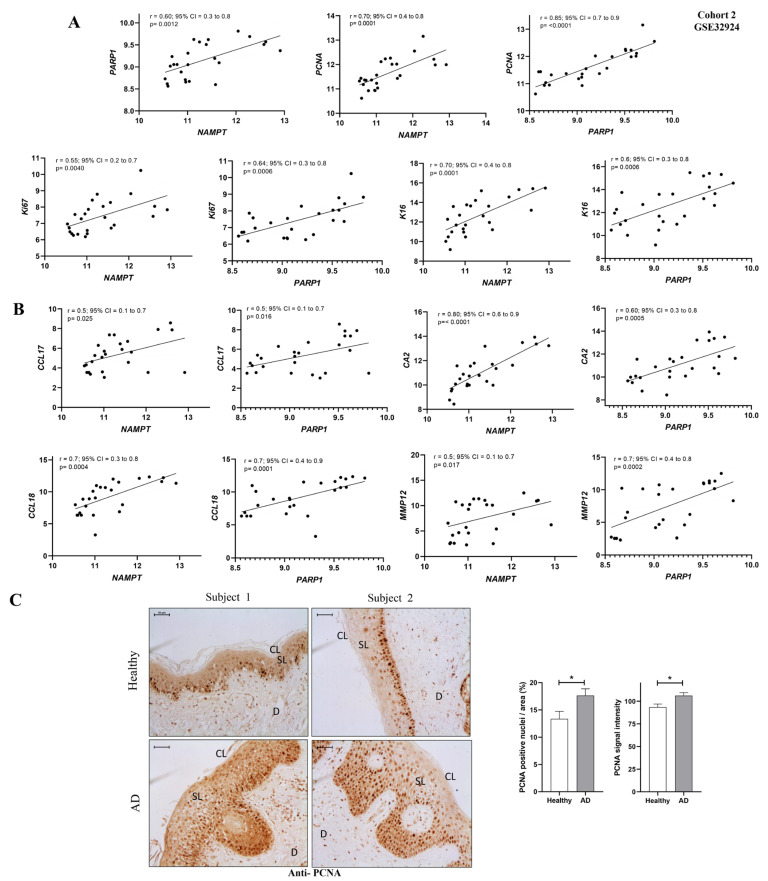
NAMPT and PARP1 levels correlated with lesional biomarkers of AD. (**A**) Correlation of *NAMPT* and *PARP1* with proliferative markers. (**B**) Correlation of *NAMPT* and *PARP1* with inflammatory markers. Univariate correlations were performed using transcriptomic data from GEO cohort 2 and calculated using Spearman’s correlation coefficient. (**C**) Representative images and analysis of biopsy sections from healthy (*n* = 10) and AD (*n* = 6) skin biopsies that have been immunostained with anti-PCNA. The mean ± SEM of each group is shown. *p* values were calculated using the nonparametric Mann–Whitney test, * *p* ≤ 0.05. Scale bar is 50 μm in all panels. CL: cornified layer; D: dermis; SL: spinous layer.

**Figure 5 ijms-24-07992-f005:**
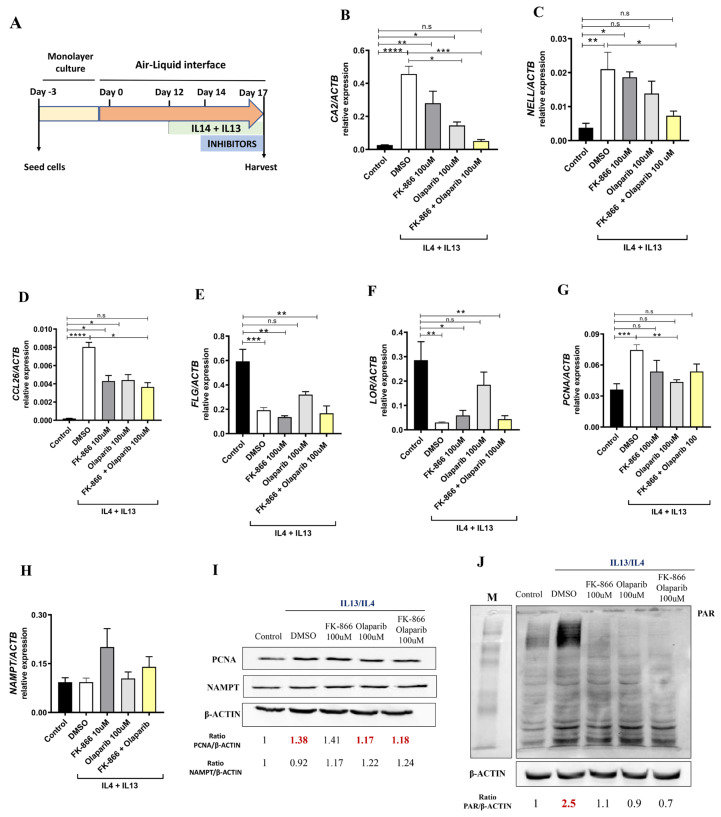
Reduction in pathology-associated biomarkers in human organotypic 3D skin model of AD. (**A**) Experimental design. Transcript levels of the indicated genes encoding markers of inflammation (**B**–**D**,**H**), epidermal differentiation (**E**,**F**) and proliferation (**G**) were determined in human organotypic 3D skin pretreated with 10 ng/mL IL13 and IL4 in the presence of vehicle (DMSO) or the indicated inhibitors. Protein levels of interest were also determined by Western blot (**I**,**J**). The mean ± SEM of each group is shown. Results are representative of 3 independent experiments. *p* values were calculated by Kruskal–Wallis test and Dunn’s multiple comparisons test. * *p* ≤ 0.05, ** *p* ≤ 0.01, *** *p* ≤ 0.001, **** *p* ≤ 0.0001. M, molecular weight markers.

**Figure 6 ijms-24-07992-f006:**
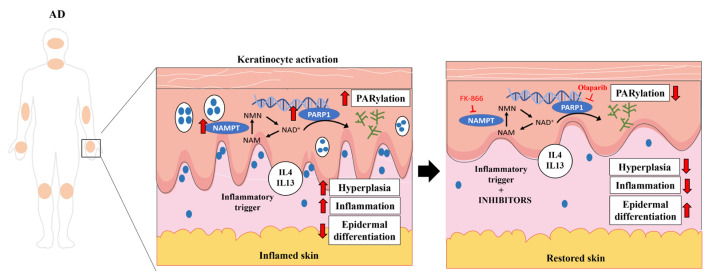
NAD^+^ and PAR metabolism is involved in AD pathogenesis. PARP1 hyperactivation in keratinocytes fueled by NAMPT-derived NAD^+^ exacerbates the inflammation and hyperplasia of AD. Consequently, pharmacological inhibition of NAMPT and PARP1 decreases disease-associated inflammation and proliferation, restoring skin homeostasis. Smartservier resources have been used to generate this figure.

## Data Availability

Data are contained within the article.

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
