# Peer review of "NAMPT and PARylation Are Involved in the Pathogenesis of Atopic Dermatitis"

_ijms, 2023, doi:10.3390/ijms24097992_

Round 1

Reviewer 1 Report

Arroyo and co-workers presented a manuscript entitled: “NAMPT and PARylation are involved in the pathogenesis of atopic dermatitis”. They reported that exist an altered metabolism of NAD+ and PAR in skin with atopic dermatitis (AD) and a strong correlation between NAMPT and PARO1 expression and lesional status of AD. Furthermore, by using a human 3D organotypic skin model, they showed that the pharmacological inhibition of NAMPT and PARP reduced AD associated inflammation and keratinocyte proliferation.

The topic presented is very interesting, and the manuscript is well written. However, several issue should be addressed.

1)      The authors should clarify the number of patients on which they performed the immunohistochemical evaluation of NAMPT, PAR and PCNA. Did they use only three and two patients, respectively? If its so, they should increase the number of patients.

2)      They showed a correlation between NAPT and PARP transcript levels and markers of epidermal hyperplasia and inflammation. Moreover, they showed that patients with elevated NAMPT and PAR levels had also an elevated expression of PCNA. Did they evaluated if the increased expression of NAMPT or PAR increased also the proliferation status of cells? Did they evaluate if the expression of NAMPT or PAR occur in the same cells with high PCNA expression?

3)      The authors use a 3D organotypic model of AD. However, I think it is important to show the histology of the models in order to check the real quality of the model used and to verify if their conclusion are correct. I think it’s important to show:

-          Histology of the “healthy” model (without the treatment with Interleukin), in order to evaluate if the epidermis is completed formed.

-          Histology of the AD model (treated with interleukin), to evaluate the AD model. In this case it should be necessary and evaluation of the epidermal thickness in AD vs healthy model.

-          Evaluate the expression level of NAMPT and PAR in AD vs healthy model.

-          Moreover, it would be important evaluate the expression of proliferation and differentiation markers in not treated vs treated AD model.

Author Response

Arroyo and co-workers presented a manuscript entitled: “NAMPT and PARylation are involved in the pathogenesis of atopic dermatitis”. They reported that exist an altered metabolism of NAD+ and PAR in skin with atopic dermatitis (AD) and a strong correlation between NAMPT and PARO1 expression and lesional status of AD. Furthermore, by using a human 3D organotypic skin model, they showed that the pharmacological inhibition of NAMPT and PARP reduced AD associated inflammation and keratinocyte proliferation.

The topic presented is very interesting, and the manuscript is well written. However, several issue should be addressed.

1)      The authors should clarify the number of patients on which they performed the immunohistochemical evaluation of NAMPT, PAR and PCNA. Did they use only three and two patients, respectively? If its so, they should increase the number of patients.

As indicated in the M&M section (line 92), skin biopsies from healthy donors (n = 10) and AD patients (n = 6) were used.

2)      They showed a correlation between NAPT and PARP transcript levels and markers of epidermal hyperplasia and inflammation. Moreover, they showed that patients with elevated NAMPT and PAR levels had also an elevated expression of PCNA. Did they evaluated if the increased expression of NAMPT or PAR increased also the proliferation status of cells? Did they evaluate if the expression of NAMPT or PAR occur in the same cells with high PCNA expression?

This is an interesting suggestion. We have now performed a double immunofluorescence for NAMPT and PCNA, and the results revealed that most NAMPT+ cells were in the spinous layer, while most PCNA+ cells were found in the basal layer. In addition, the morphological analysis also revealed that there is not a statistically significant colocalization. These results were included in Supplemental Figure 3.

3)      The authors use a 3D organotypic model of AD. However, I think it is important to show the histology of the models in order to check the real quality of the model used and to verify if their conclusion are correct. I think it’s important to show:

-          Histology of the “healthy” model (without the treatment with Interleukin), in order to evaluate if the epidermis is completed formed.

-          Histology of the AD model (treated with interleukin), to evaluate the AD model. In this case it should be necessary and evaluation of the epidermal thickness in AD vs healthy model.

-          Evaluate the expression level of NAMPT and PAR in AD vs healthy model.

-          Moreover, it would be important evaluate the expression of proliferation and differentiation markers in not treated vs treated AD model.

We agree that this histological analysis will be informative. However, this 3D organotypic model was already validated with histology (reference 14) and it is not possible for us to perform more experiments since we have experienced many problems with the supply of the inserts required to these cultures after the pandemic.

Reviewer 2 Report

The present Research Article, ijms-2229164 entitled: (NAMPT and PARylation are involved in the pathogenesis of atopic dermatitis)

The current research article contained a good work, good analysis and written in a good way. However, I have some minor comments as following:

- In page (4) in section 2.1; I recommend to add inclusion and exclusion criteria of patients 

-In page (4) in section 2.2; how the samples were processed 

-In page (15) please add magnification –

Change format of p- value to italic form in in the manuscript.

-Please double check the key of p-values of  **** in figure(1, 2, 3, 4) it refer to ???

Author Response

The present Research Article, ijms-2229164 entitled: (NAMPT and PARylation are involved in the pathogenesis of atopic dermatitis)

The current research article contained a good work, good analysis and written in a good way. However, I have some minor comments as following:

- In page (4) in section 2.1; I recommend to add inclusion and exclusion criteria of patients 

We have included these criteria, as requested.

-In page (4) in section 2.2; how the samples were processed

Data were retrieved, plotted and statistical analysis performed as indicated in section 2.6.

-In page (15) please add magnification –

All images have a scale bar.

Change format of p- value to italic form in in the manuscript.

Done.

-Please double check the key of p-values of  **** in figure(1, 2, 3, 4) it refer to ???

This has been corrected.

Reviewer 3 Report

In this paper, Arroyo et al demonstrated the critical role of NAD+ and poly (ADP-ribose) (PAR) metabolism in oxidative stress and skin inflammation. They found that hyperactivation of PARP1 in response to DNA damage induced by reactive oxygen species, and fuelled by NAMPT-derived NAD+, mediated inflammation through parthanatos cell death in preclinical models of psoriasis in zebrafish and human organotypic 3D skin models of psoriasis. They report altered NAD+ and PAR metabolism in the skin of AD patients and a strong correlation between NAMPT and PARP1 expression and lesional status of AD. They also suggest new potential treatments for AD patients. This is very dreamy work, and it is hoped that future development of therapeutic agents, etc. will be possible.

It is suggested that oxidative stress and DNA damage response are deeply involved in inflammation in AD and psoriasis, and the development of novel therapeutic agents targeting these factors is desirable.

Throughout, the data were analyzed from multiple perspectives, including immunostaining data, analysis of mRNA expression and protein levels, and western blotting analysis. The results also show consistent data, strongly suggesting that these pathways are deeply involved in AD and psoriasis.

The schema in Figure 6 also summarizes the results of this study in a straightforward manner and is very easy to understand.

I have read it carefully and have nothing to add regarding the content of the experiment.

One question I have about the whole process is related to NMN (nicotinamide mononucleotide), which I understand is a substance that is expected to have rejuvenating and other effects, but could this mean that it could worsen inflammation such as AD according to the schema in Figure 6? Does this mean that it may exacerbate inflammation such as AD according to the schema in Figure 6? I ask this question because I think this is a very important point as the substance is being applied clinically. I would appreciate it if you could answer as much as you know.

Author Response

In this paper, Arroyo et al demonstrated the critical role of NAD+ and poly (ADP-ribose) (PAR) metabolism in oxidative stress and skin inflammation. They found that hyperactivation of PARP1 in response to DNA damage induced by reactive oxygen species, and fuelled by NAMPT-derived NAD+, mediated inflammation through parthanatos cell death in preclinical models of psoriasis in zebrafish and human organotypic 3D skin models of psoriasis. They report altered NAD+ and PAR metabolism in the skin of AD patients and a strong correlation between NAMPT and PARP1 expression and lesional status of AD. They also suggest new potential treatments for AD patients. This is very dreamy work, and it is hoped that future development of therapeutic agents, etc. will be possible.

It is suggested that oxidative stress and DNA damage response are deeply involved in inflammation in AD and psoriasis, and the development of novel therapeutic agents targeting these factors is desirable.

Throughout, the data were analyzed from multiple perspectives, including immunostaining data, analysis of mRNA expression and protein levels, and western blotting analysis. The results also show consistent data, strongly suggesting that these pathways are deeply involved in AD and psoriasis.

The schema in Figure 6 also summarizes the results of this study in a straightforward manner and is very easy to understand.

I have read it carefully and have nothing to add regarding the content of the experiment.

We are pleased with the reviewer’s comments on our manuscript.

One question I have about the whole process is related to NMN (nicotinamide mononucleotide), which I understand is a substance that is expected to have rejuvenating and other effects, but could this mean that it could worsen inflammation such as AD according to the schema in Figure 6? Does this mean that it may exacerbate inflammation such as AD according to the schema in Figure 6? I ask this question because I think this is a very important point as the substance is being applied clinically. I would appreciate it if you could answer as much as you know.

This is a very important point, and, in fact, this is our conclusion. NAD boosters have shown to rejuvenate and/or delay aging in mice models, and this makes sense since NAD+ levels decline in several tissues, including the skin, with aging. However, the results from the present study and a previous one published in Plos Biology about psoriasis strongly suggest that NAD+ booster may be detrimental for chronic skin inflammatory disorders.

Round 2

Reviewer 1 Report

I appreciate the work made by the authors, and the manuscript is now improved. However, they should discuss the results of the no co-localization of NAMPT and PCNA. Why most NAMPT+ cells are in the spinous layer, while most PCNA+ cells are in the basal layer? Do they think that the overexpression of NAMPT is a consequence of the hyperproliferation state of keratinocytes in the basal layer? 

Author Response

This is an interesting point. It has been shown in mouse model of acute inflammation (PMID 17579037, 20975043) that NAD+ can be release, so it is likely that NAMPT+ keratinocytes can release NAD+ to the extracellular space and sustain the high proliferation rate in the whole epidermis. This has now been discussed in the revised version (lines 295-298).
